# Two typical merging events of oceanic mesoscale anticyclonic eddies

Zi-Fei Wang[1], Liang Sun[1], Qiu-Yang Li[1], Hao Cheng[1]

[1] School of Earth and Space Sciences, University of Science and Technology of China, Hefei, Anhui, 230026, PR China

*Correspondence to*: Liang Sun (sunl@ustc.edu.cn)

**Abstract.** The long-term theoretical "energy paradox" of whether the final state of two merging anticyclones contains more energy than the initial state is studied by considering two typical merging events of ocean mesoscale eddies. The results demonstrate that the total mass (volume), total circulation (area integration of vorticity), and total angular momentum (AM) are conserved if the orbital AM relative to the center of mass is taken into account as the eddies rotate around the center of mass before merging. For subsurface merging, the mass trapped by the Taylor–Proudman effect above the subsurface eddies
should also be included. Both conservation laws of circulation and orbital AM have been overlooked in previous theoretical studies. The total eddy kinetic energy decreases slightly after merging due to fusion. On the contrary, the total eddy potential energy (EPE) increases significantly after merging. The increase of EPE is mostly supported by the loss of gravitational potential energy (PE) via eddy sinking below the original level. This implies that the merging of eddies requires that background gravitational PE convert to EPE. In contrast, the vorticity and enstrophy consequently decrease after merging.
Thus, the eddy merging effect behaves as a "large-scale energy pump" in an inverse energy cascade. It is noted that eddy conservation and conversion laws depend on the laws of physical dynamics, even if additional degrees of freedom can be provided in a mathematical model.

## 1 Introduction

Mesoscale eddies, i.e., coherent vortexes with a rotational core, usually have a long life cycle of weeks or months and carry
long-distance transports of heat, salt, and other passive tracers [Chelton et al., 2011a; Dong et al., 2014; McGillicuddy et al., 2011; Zhang et al., 2014; Bosse et al., 2019] by trapping those tracers along with the water [Xu et al., 2014; Torres et al., 2018]. During the lifetime of an eddy, complex dynamic processes, such as merging and splitting, which are associated with eddy genesis and termination, often occur. This in turn modulates the eddy's life cycle and transports. In addition, there are incoherent eddies, which typically do not have a core and do not have a well-defined eddy radius. These incoherent eddies are
also important because they contain most of the eddy kinetic energy (EKE) in the ocean is from incoherent eddies [Torres et al., 2018] and are responsible for most of the eddy transport of tracers is mostly due to incoherent motions [Su et al., 2018; Zhang et al., 2019].

There is a long-term unresolved "paradox" pertaining to the merging of two like-signed oceanic mesoscale eddies, i.e., whether the merged eddy has more or less energy than the sum of the two original eddies. This paradox first emerged from a theoretical

study on the merging of two anticyclonic, zero-potential-vorticity plane eddies by Gill and Griffiths [Nof and Simon, 1987; Cushman-Roisin, 1989; Lumpkin et al., 2000]. According to the theory, if mass and potential vorticity are conserved by two anticyclones, the final state should contain more energy than the initial state, which implies that an additional amount of energy must be supplied to complete the process. This study seemed to open a "Pandora's box", as subsequent studies were in contradiction about which eddy properties (mass, potential vorticity, energy, angular momentum) should be conserved after

merging. The conservation of mass is a generally accepted assumption and has been validated in experiments [Nof and Simon, 1987]. However, there are still some merger scenarios in which mass is not conserved [Cushman-Roisin, 1989; Lumpkin et al., 2000]. Similarly, the conservation of potential vorticity (PV) [Gill and Griffiths 1981; Cushman-Roisin 1989; Nof, 1990] has been abandoned in some studies [Griffiths and Hopfinger, 1987; Nof and Simon, 1987; Nof, 1988; Lumpkin et al., 2000]. In contrast, the previously overlooked conservation of angular momentum (AM) [Cushman-Roisin, 1989] has become

generally accepted in various models [Nof, 1990; Pavia and Cushman-Roisin, 1990; Lumpkin et al., 2000]. Consequently, in theoretical scenarios of merging, the merged eddy might have less [Lumpkin et al., 2000], the same [Nof and Simon, 1987; Pavia and Cushman-Roisin, 1990; Lumpkin et al., 2000], or more energy [Griffiths and Hopfinger 1987] than before merging, depending on the assumptions. Because these eddy models have only two parameters (amplitude and radius), there is the dilemma of the parameters being less numerous than the conservation laws (e.g., mass, vorticity, momentum, energy) [Pavia

and Cushman-Roisin, 1990; Lumpkin et al., 2000]. This leads again to the question of which conservation laws should be applied to eddy merger [Lumpkin et al., 2000]. One possible way to solve this dilemma is to use a complex eddy model with more parameters, with which more conservation laws might be held simultaneously.

In addition, an effective way to dispel this "paradox" in eddy merger scenarios is to use oceanic observations to examine the above assumptions. However, this has seldom been studied because the field observations by research cruises and floats can

hardly capture eddy merger events, except in only a few cases [Cresswell, 1982; Sangra et al., 2005; Raj et al., 2016]. On the other hand, satellite observations provide a different way of observing eddy motions. Based on sea level anomaly (SLA) data, the Genealogical Evolution Model (GEM), an efficient logical model, was developed to track the dynamic evolution (merging and splitting) of eddies [Li et al., 2014; Li et al., 2016].

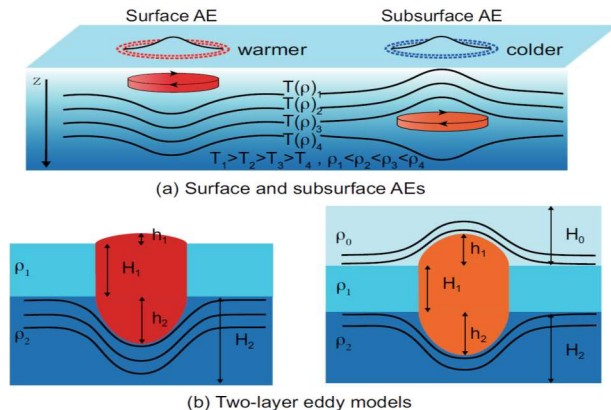

Figure 1. (a) Surface and subsurface anticyclonic eddies (AEs) and (b) two-layer eddy models used in the present study.

The motivation of this study was to test the conservation laws without any prior assumptions of conservation by calculating the eddy properties. To this end, we firstly chose two typical merger events by using GEM. Next, the eddies were distinguished as surface or subsurface eddies (Fig. 1a) according to sea surface temperature (SST) data [Assassi et al., 2016]. Secondly, we used a vertical two-layer model (Fig. 1b) [Lumpkin et al., 2000] and a Gaussian model for horizontal shape according to observations [Wang et al., 2015, Yi et al., 2015]. There were many more eddy parameters than potential conservation laws, which avoided the aforementioned dilemma. Thirdly, we estimated pre- and post-merging eddy parameters (Fig. 1a, b) with a nonlinear optimal fitting approach [Wang et al., 2015]. Finally, we calculated the eddy properties to determine what occurred after the two typical eddy mergers.

The paper is structured as follows. A brief description of the data, eddy identification, eddy parameters, and eddy properties is presented in Section 2. In Section 3, the merging processes of surface AEs and subsurface AEs are both described and then the conservation and conversion laws are tested by using the eddy properties. The sensitivity of the result to the parameters is discussed in Section 4. Finally, the conclusions are presented in Section 5.

## 2. Data and Method

### 2.1 Data

The sea level anomaly (SLA) data used here were from the merged and gridded satellite product of MSLA (Maps of SLA), which is produced and distributed by AVISO (http://www. aviso.oceanobs.com/) based on TOPEX/Poseidon, Jason-1, ERS-1, and ERS-2 data [Ducet et al., 2000]. Currently, the products are available on a daily scale with $0.25° \times 0.25°$ resolution for the global ocean as DUACS DT14 [Pujol et al., 2016]. It is worth pointing out that such resolution applies only to the data resolution, not the physical signal resolution, due to AVISO's low resolution of 100–200 km [Ducet et al., 2000; Chelton et al., 2011a; Amores et al., 2018; Ballarotta et al., 2019].

The SST data were produced by the Operational Sea Surface Temperature and Sea Ice Analysis (OSTIA) system [Donlon et al., 2011] and downloaded at the Asia-Pacific Data-Research Center (APDRC http://apdrc.soest.hawaii.edu/data/). The analysis of SST data has global coverage with a spatial resolution of $0.05° \times 0.05°$ and a temporal resolution of 1 day. The ocean vertical density profile data were from the NCEP Global Ocean Data Assimilation System (GODAS) [Behringer et al., 2004] and were downloaded from NOAA (https://www.esrl.noaa.gov/psd/data/gridded/data.godas.html).

Eddy merger events were tracked using the GEM model [Li and Sun, 2015; Li et al, 2016], which is an efficient logical model for tracking the dynamic evolution of mesoscale eddies (merging and splitting) from satellite SLA data.

### 2.2 Identification of surface and subsurface anticyclonic eddies

Surface eddies are distinguished from subsurface eddies by whether their core is in the surface layer or located inside the water column (Fig. 1a). Consequently, the surface anticyclonic eddies (AEs) have positive SST anomalies (SSTAs) and the

subsurface AEs have negative SSTAs [Assassi et al, 2016]. Here, we used the method proposed by Assassi et al. (2016) to identify surface AEs (SSTA>0) and subsurface AEs (SSTA<0). If the SSTA index > 0, then the eddy was identified as a surface AE. Otherwise, if the SSTA index < 0, then the eddy was identified as a subsurface AE. The SSTA was obtained by
removing the background SST, which is the weighted average of the SST within the 4.5° × 4.5° box, as used previously [Assassi et al., 2016].

**2.3 Eddy parameters**

In this study, we used a Gaussian model for SLAs, which is the most typical model for ocean mesoscale eddies [Wang et al., 2015; Yang et al., 2019], to obtain the eddy properties. The SLA field h($x,y$) with several adjacent eddies before merging can
be expressed as [Yi et al, 2015; Chen et al., 2019]

$$h(x,y) = b + \sum_{i=1}^{n} A_i \times \exp[-\frac{(x-x_{0i})^2}{2L_{xi}^2} - \frac{(y-y_{0i})^2}{2L_{yi}^2}] \qquad (1)$$

where $x$, $y$ are the zonal and meridional coordinates, respectively; $x_0$, $y_0$ is the position of the eddy center; $L_x$ and $L_y$ correspond to the longitude and latitude radii, respectively; and $A$ denotes the eddy amplitude. The eddy parameters ($A$, $L_x$, and $L_y$) are obtained by using nonlinear fitting [Wang et al., 2015]. The eddy size, i.e., eddy area S = $\pi L_x L_y$, is calculated by integrating
an ellipse. It is useful to point out that eddy area is an extensive quantity; this is the reason why the eddy area is the intrinsic parameter in estimating eddy viscosity [Li et al., 2018]. Next, the velocity ($u,v$) and vorticity ($\xi$) of the eddy field are calculated by geostrophic approximation (if the x- and y-axis origins are at the eddy center).

$$u = \frac{gAy \times \exp(-\frac{x^2}{2L_x^2} - \frac{y^2}{2L_y^2})}{fL_y^2} \qquad (2a)$$

$$v = \frac{gAx \times \exp(-\frac{x^2}{2L_x^2} - \frac{y^2}{2L_y^2})}{-fL_x^2} \qquad (2b)$$

$$\xi = \frac{gA \times \exp(-\frac{x^2}{2L_x^2} - \frac{y^2}{2L_y^2})}{f}(\frac{x^2}{L_x^4} + \frac{y^2}{L_y^4} - \frac{1}{L_x^2} - \frac{1}{L_y^2}) \qquad (2c)$$

The parameter $b$ represents the vertical eddy shift, which is critical for properly composing and fitting the eddy parameters [Wang et al., 2015]. In addition, in this study, we point out for the first time that it is associated with the PE conversion balance. For a two-layer model, as shown in Fig. 1b, the upper (lower) layer has a thickness of $H_1$ ($H_2$) and a density of $\rho_1$ ($\rho_2$). The surface AE consists of three parts—the upper surface $h_1 = A$, the lower surface $h_2 = \frac{\rho_1}{\rho_2-\rho_1}A$, and the eddy body of height
$H_1$. Typically, we have $h_1 << h_2 << H_1$. If both $h_1$ and $h_2$ are very small and can be ignored, the model becomes a one-layer model of a cylinder [Sangra et al., 2005] or a plane model, as used in many previous theoretical models. On the other hand, if $H_1$ is too small to be ignored, it becomes a lens model [e.g., Lumpkin et al., 2000]. For subsurface AEs, there is an additional surface layer of thickness $H_0$ and density $\rho_0$ over the eddy (Fig. 1b). The upper surface $h_1 = \frac{\rho_1}{\rho_1-\rho_0}A$ and the lower surface $h_2 = \frac{\rho_1}{\rho_2-\rho_1}A$ satisfy $h_1 \sim h_2 << H_1$. In the present study, both $H_0$ and $H_1$ were chosen to be 200 m, partly according to some

recent observations [Zhang et al., 2015; Bashmachnikov 2017; Li et al., 2017; Wang, 2017; Mason et al., 2019]. The sensitivity of the result to choice of depth is discussed in Section 4.

## 2.4 Eddy properties

The eddy properties are calculated by integration of the proper parameters within eddy. The originally identified boundary of eddy may lead to unexpected sharp decreasing of eddy properties before merger [e.g., Laxenaire et al, 2018; Cui et al., 2019]. This is mainly due to that the previous eddy detection methods lack proper segmentation algorithm, which was illustrated in Li and Sun (2015). In this study, the integration area is an ellipse with major axis and minor axis of $2L_x$ and $2L_y$, respectively. As the density varies little from surface to deep sea, the anticyclone mass is calculated by numerical integration of volume:

$$V = \iint (H_1 + h_1 + h_2)dxdy \tag{3}$$

where $H_1$ is the depth of the vortex body layer and $h_1$ and $h_2$ are the upper and lower interface anomalies, respectively, as shown in Fig. 1b. The relative eddy circulation $\Gamma$ is calculated by surface integration of PV (Gill A.E., 1982, p 192),

$$\xi - f\frac{h_1+h_2}{H_1} \tag{4}$$

or volume integration of the PV anomaly (Gill A.E., 1982, p 192):

$$\Gamma = \iint \left(\xi - f\frac{h_1+h_2}{H_1}\right)dxdy = \iint \left(\frac{f+\xi}{H_1+h_1+h_2} - \frac{f}{H_1}\right)(H_1 + h_1 + h_2)dxdy \tag{5}$$

where $f$ is the Coriolis parameter. The relative PV anomaly mean $\xi_m = \Gamma/S$ is the area average of circulation $\Gamma$. The relative AM is calculated by integration of torque, where the x- and y-axis origins are at the eddy center:

$$L = \iint (vx - uy)(H_1 + h_1 + h_2)dxdy \tag{6}$$

The eddy kinetic energy (EKE) per mass is calculated by integration as follows:

$$E_k = \iint \left(\frac{u^2+v^2}{2}\right)(H_1 + h_1 + h_2)dxdy \tag{7}$$

The eddy potential energy (EPE) consists of the effective PE of the upper interface and the lower interface, as $H_1$ does not change during the merging process [e.g., Lumpkin et al., 2000]:

$$E_p = \iint \left(\frac{1}{2}g_1'h_1^2 + \frac{1}{2}g_2'h_2^2\right)dxdy \tag{8}$$

where $g_1' = \frac{\rho_1-\rho_0}{\rho_1}g$ and $g_2' = \frac{\rho_2-\rho_1}{\rho_1}g$ are the reduced gravity. The eddy enstrophy is calculated by integration:

$$E_s = \iint \left(\frac{1}{2}\xi^2\right)dxdy \tag{9}$$

The eddy gravitational PE referring to background sea level with eddy shift parameter $b$ is calculated as follows:

$$E_g = \iint \rho_1 gb(H_1 + h_1 + h_2)dxdy \tag{10}$$

## 2.5 Error estimation

Since the above eddy properties and parameters are cacluated from SLA field, we need to estimate the errors of the values from the calculation. We first estimate the errors of eddy parameters (e.g., $A$, $Lx$, and $Ly$) obtained by nonlinear fitting. This is

simple, because the outputs of the fitting algorithm (Wang et al., 2015) have already included the standard deviation (e.g., $\delta A$,
$\delta b$ and $\delta S$) of each parameter, and the coefficient of determination ($R^2$). Typically, the standard deviations are 2~8% of eddy parameters, and the fitting performance $R^2$ is from 0.87 to 0.98 in this study.

Secondly, we estimate the standard deviations of eddy properties. Since we have used numerical integration of eddy parameters to obtain eddy properties, there are no simple and explicit relations between eddy properties and eddy parameters. The exact standard deviations of eddy properties can hardly be obtained in this way. Here we approximately estimate the standard deviations of eddy properties by assuming that eddy is a circle with same area $S$ of original ellipse. Then the eddy properties can be expressed as functions of eddy parameters (e.g. $A$ and $S$) after integration. The standard deviations of eddy properties can now be estimated with standard deviations of eddy parameters. For example, the eddy enstrophy in Eq. (9) is $E_s=cA^2/S$, where $c$ is the integration constant. Then the standard deviations of eddy enstrophy is $\delta E_s = E_s(2\delta A/A+\delta S/S)$. We use these standard deviations to draw errorbars in figures.

## 3. Results

### 3.1 Merging of surface AEs

The merging event consists of two AEs (AE1 and AE2) south of the Kuroshio Extension within 17° to 22° N and 180° to 174° W from May 19 to June 10, 2013 (Fig. 2). Before the merging, AE1 was located at approximately 19° N and 178° W and AE2 was located to its northeast (Fig. 2a). As both eddies had positive SSTAs, they were surface AEs, as discussed in Section 2. Both eddies then moved westward from 178° W to 179° W (Fig. 2d). During this time, AE1 and AE2 co-rotated anticyclonically. This co-rotation was also observed by Li et al. (2016). Next, AE1 and AE2 gradually merged into a larger AE on June 1 (Fig. 2c). In this case, the upper (lower) layer, ranging from 0 to 200 m (200 to 1000 m) depth, had a mean density of 1024.2 (1029.4) kg/m$^3$, according to the GODAS data for this region. The height of the eddy body $H_1$ was 200 m, and the lower surface height $h_2$ (27.3 m) was approximately 195 times that of the eddy amplitude $h_1$ (0.14 m).

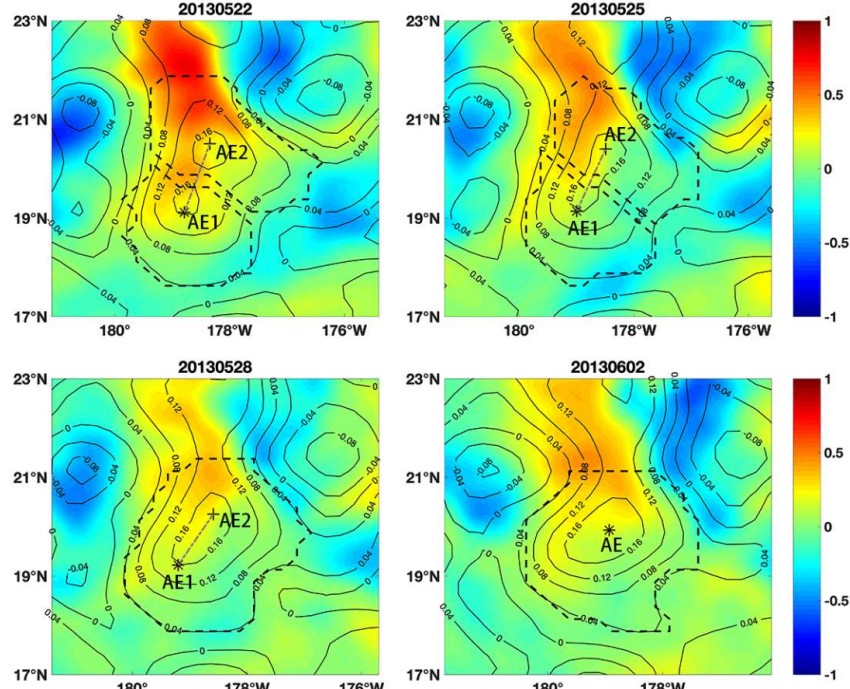

Figure 2. Merging event of surface anticyclonic eddies. The colors and contours represent SST and SSHA values, respectively. The eddy boundaries are labeled with dashed curves. The cross and asterisk represent eddy centers identified by the fitting method. The dashed line connecting eddy centers rotates anticyclonically during the merging event.

To illustrate clearly how the eddies changed during the merging process, we first calculated the parameters of both eddies. It is obvious that both eddies had positive SSTAs and that the merged AE also had positive SSTAs (Fig. 3a). They all were surface AEs. The fitted eddy parameters are shown in Fig. 3a. The first fitted parameter is the vertical shift $b$, representing the background SLA. In this case, $b$ was very small, and it decreased a small amount from approximately 0.045 m before the merging to 0.038 m after the merging. The second parameter is the amplitude of the eddy. Before merging, the amplitude of AE2 increased gradually from 0.12 m to 0.14 m and, notably, the amplitude of AE1 decreased gradually from 0.12 m to 0.08 m. After merging, the amplitude of the merged eddy increased continually from 0.16 m to 0.17 m. In contrast, the area of each eddy seldom changed before the merging. AE2 had a large area of $1.7 \times 10^4$ km$^2$, and AE1 had a smaller area of $0.9 \times 10^4$ km$^2$.

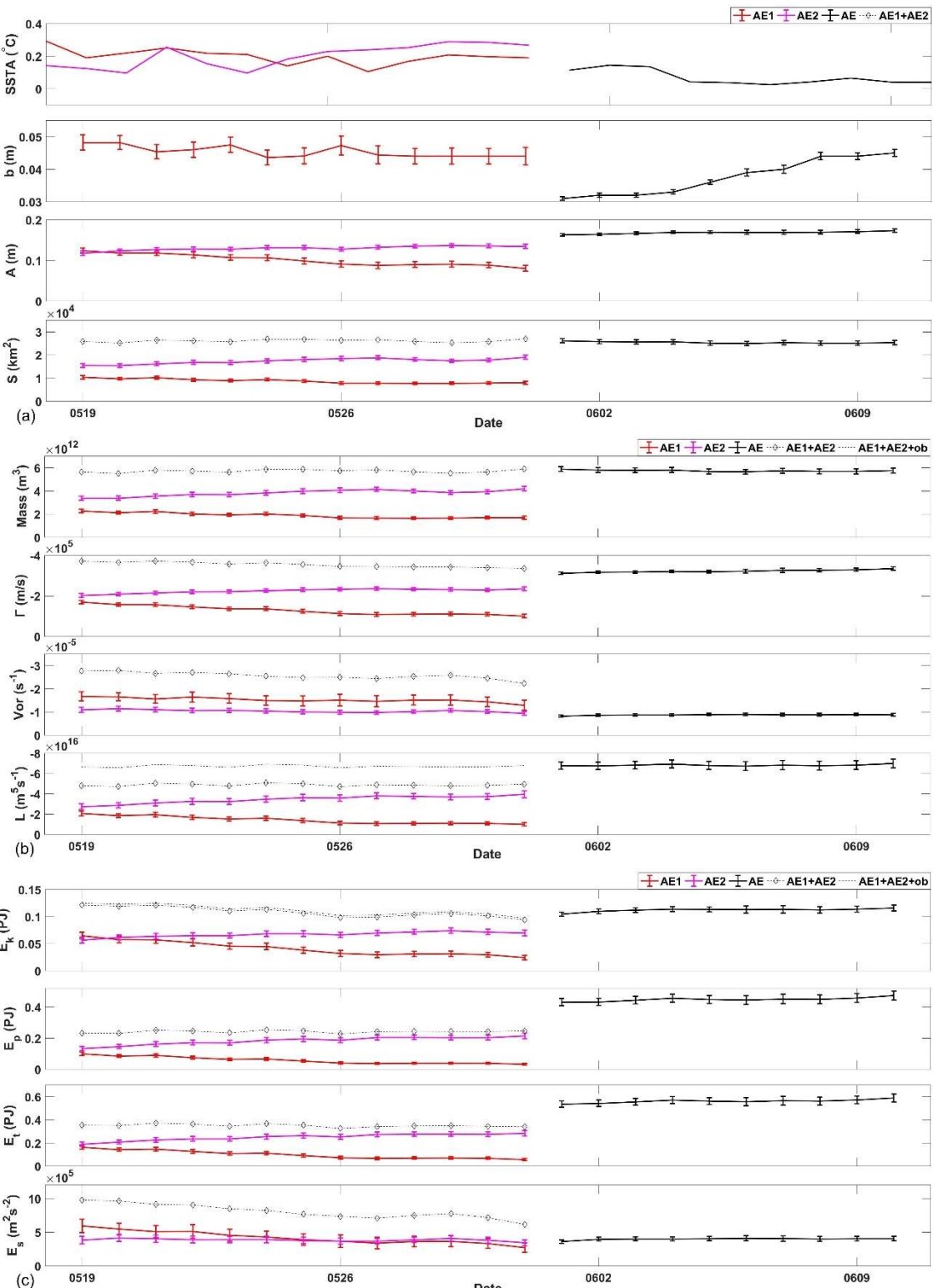

Figure 3. The parameters in the merging event of the surface mesoscale eddies, where "AE1," "AE2," and "AE" represent the eddies, respectively, and "ob" represents the obtial AM assocated with co-rotation. The error bars indicate the standard deviation of the value. (a) SSTA, background parameter, eddy amplitude, and eddy area; (b) mass, circulation, vorticity, and angular momentum; and (c) EKE, EPE, total mechanical energy, and enstrophy.

Next, we calculated the eddy properties using the above parameters. The mass (volume) of the eddies experienced changes similar to those of the eddy area (Fig. 3b). This occurred because $H_1$ dominates the whole depth as $h_1 \ll h_2 \ll H_1$. The relative PV was averaged within each eddy. As shown in Fig. 3b, the vorticity of AE2 varied by a very small amount and the vorticity of AE1 decreased gradually before merging. It is noted that the vorticity of AE2 was significantly smaller, although it had a

larger amplitude, which was obtained from the SLA field according to Eq. (1). This is because vorticity is not only proportional to eddy amplitude but is also inversely proportional to eddy area. After the merging, the vorticity of the merged eddy became very small, significantly smaller than that of AE2. The calculated circulation and AM of the eddy are shown in Fig. 3b. The results are similar to those for the amplitude, i.e., a larger amplitude with larger circulation and larger AM.

Finally, we calculated the energies of the eddies. Both the EKE and EPE had similar variations before the merging. However,

they were quite different after the merging: EKE decreased but EPE increased. This point is addressed in detail in the following subsection.

## 3.2 Merging of subsurface AEs

The second typical horizontal merging process was between two subsurface anticyclones. It occurred in the period from August 29 to September 22, 2015, in the region between 18° and 23° north latitude and 179° east longitude and 175° west longitude,

roughly to the west of the surface vortex merging event described in the previous section. In this case, three layers were divided as 0–200 m, 200–400 m, and 400–1000 m, respectively. The mean densities of the layers were 1024.2, 1028.0, and 1030.8 kg/m³, respectively, according to the GODAS data for the regions. The height of the eddy body $H_1$ was 200 m, and the upper (lower) surface height $h_1$ ($h_2$) was 104 m (98.4 m), approximately 348 (328) times the eddy amplitude.

At the beginning of the merging, anticyclone AE1 was located at approximately 20° N latitude and 177° W longitude, and

AE2 was northwestward alongside AE1. Then, AE1 and AE2 approached each other with a clockwise rotation and eventually merged into AE on September 13. Similarly, Fig. 4 shows a series of snapshots of this subsurface merging event. It can be seen that there is a clear cold core filled by negative SSTA in each of AE1, AE2, and AE at the beginning (top two subfigures) and ending (bottom two subfigures) stages of the process, representing a distinct anticyclonic surface vortex signal. We also noticed that during the middle of the process, on September 5, the cold core structures were interrupted by positive SSTA

areas, a small area between AE1 and AE2 and a large area, with a diameter of approximately 0.5°, west of AE1, until the two anticyclones merged. On September 14, the cold core structure of the merged vortex AE was clear again. This may have been caused by the intense mixing effect in the eddy–eddy interactions, especially when the two like-sign vortices came close to

each other. However, the type of vortex should be determined by the relatively stable continuous stage over a period of time before and after the merging, so this sudden change would not affect the identification of subsurface AE in this case.

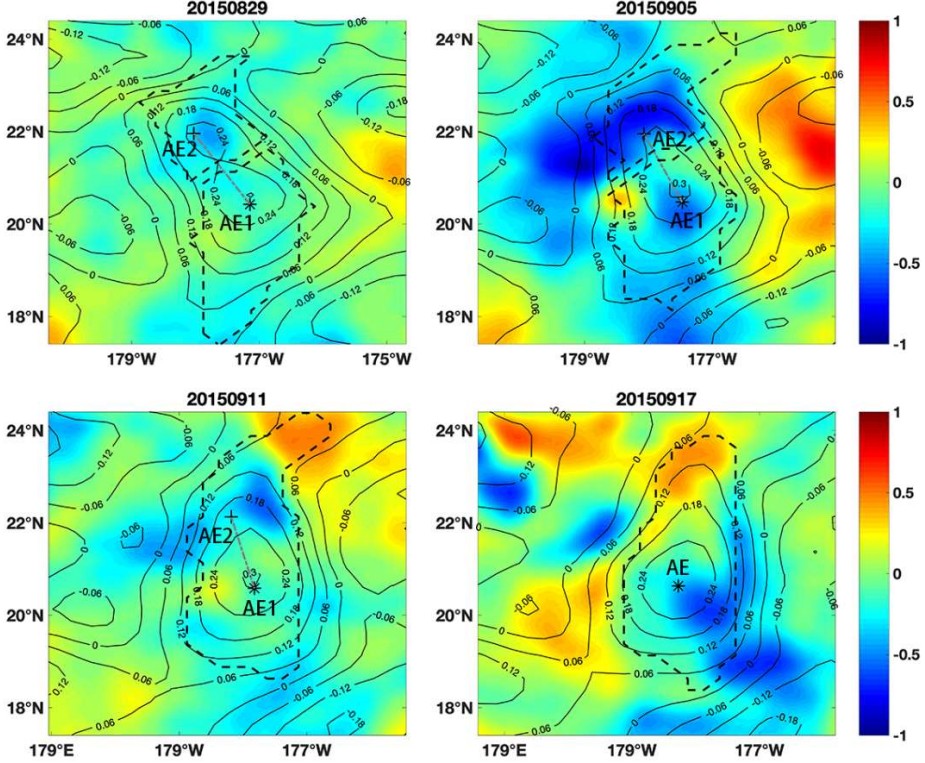

Figure 4. The merging event of subsurface anticyclonic eddies.

To illustrate clearly what occurred for the eddies during the merging process, we first calculated both eddy parameters. It is obvious that the surface eddy had a positive SSTA, but the subsurface eddy had a negative SSTA. The merged AE had a positive SSTA (Fig. 5a), i.e., the surface eddy covered the subsurface eddy. The fitted eddy parameters are shown in Fig. 5a. The first fitted parameter is the vertical shift $b$, representing the background SLA. In this case, $b$ was very small and decreased by a small amount, from approximately 0.020 m before merging to 0.001 m after merging. The second parameter is the amplitude of the eddy. Before merging, the amplitude of AE1 increased gradually from 0.28 m to 0.30 m, while the amplitude of AE2 decreased notably from 0.22 m to 0.14 m. After merging, the merged eddy had an amplitude of 0.30 m, similar to that of AE1. In contrast, the areas of the two eddies seldom changed before merging. AE1 had a large area of $2.39 \times 10^4$ km$^2$, while AE2 had a smaller area of $1.18 \times 10^4$ km$^2$.

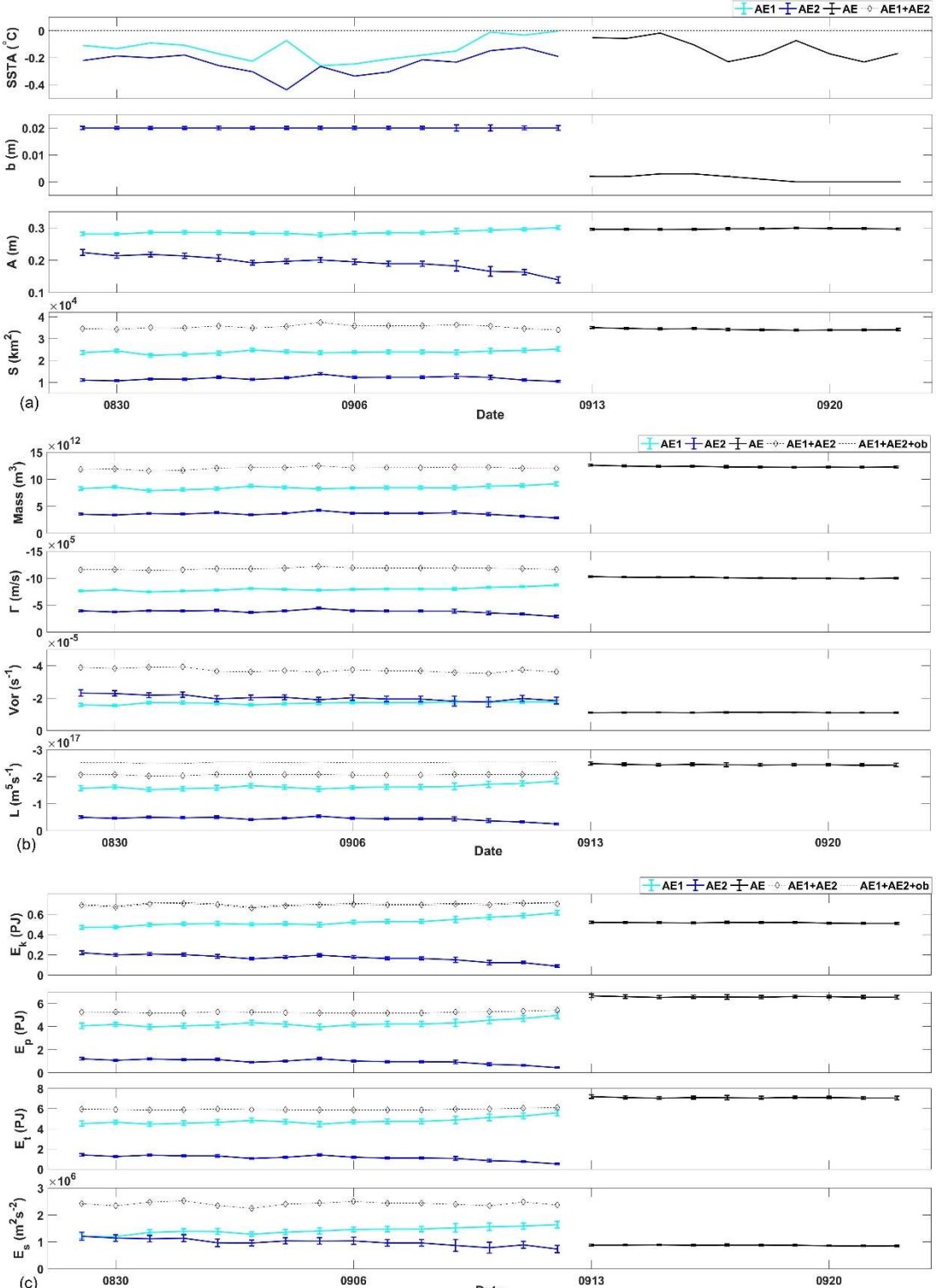

Figure 5. Same parameters as shown in Figure 3, but for subsurface eddies.

Next, we calculated the eddy properties using the above parameters. The mass (volume) of the eddies experienced changes that were similar to those of the eddy area (Fig. 5b). This occurred because $H_1$ dominates the whole depth, although $h_1$ is of the same order as $h_2$. The relative PV was averaged within each eddy. As shown in Fig. 5b, the vorticity of AE1 varied by a very small amount and the vorticity of AE2 decreased gradually before merging. It is noted that the vorticity of AE2 was significantly smaller, although it had a larger amplitude. This is because vorticity is not only proportional to amplitude but also inversely proportional to area. After merging, the vorticity of the merged eddy became very small, significantly smaller than that of AE2. Next, the circulation and AM of the eddy were calculated, as shown in Fig. 5b. The results are similar to those for the amplitude—larger amplitude with larger circulation and larger AM.

Finally, we calculated the energies of the eddies. Both the EKE and EPE had similar variations before merging. However, they were quite different after merging: EKE decreased but EPE increased. This is addressed in detail in the following subsection.

## 3.3 Conservation and conversion laws

First, we examined whether the total volume (mass) of the eddies was conserved. In the first case, the total volumes pre- and post-merging were $5.71\pm0.1 \times 10^{12}$ m$^3$ and $5.75\pm0.07 \times 10^{12}$ m$^3$, respectively, where the plus/minus amounts are standard deviations of the values. In the second case, the total volumes pre- and post-merging were $1.204\pm0.02 \times 10^{13}$ m$^3$ and $1.234\pm0.01 \times 10^{13}$ m$^3$, respectively. As shown in Figs. 3a and 5a, the total volume seldom changed in both cases. It is obvious that the merging events obeyed the law of conservation of mass.

The second conservation law is conservation of total circulation. In both cases, the total circulation of the eddies seldom changed (Fig. 3b and Fig. 5b). If circulation is conserved, the flow is referred to as circulation-preserving flow [Wu et al., 2006]. Circulation-preserving flow has minimum total enstrophy and minimum dissipation according to the Helmholtz–Rayleigh minimum dissipation theorem. The conservation of total circulation provides a method for calculating the vorticity of the merged eddy, because single eddy PV is not conserved in eddy merging events.

The third conservation law is for total AM. In the first case, the total AM before merging was $4.877\pm0.14 \times 10^{16}$ m$^5$/s, while the merged AM was $6.832\pm0.08 \times 10^{16}$ m$^5$/s. The merged AM is significantly larger than the pre-merging value. Thus, the merged eddy should have some additional sources of AM, which were ignored in previous studies. One possible source of the missing AM is orbital AM; both eddies co-rotated around the center of mass with an angular speed $\omega$ of $-1.9°$/day, as mentioned previously. This co-rotation provided additional AM of approximately $1.841 \times 10^{16}$ m$^5$/s, or 38% of the AM in both eddies. After accounting for the orbital AM contributed to the system, the total AM is approximately $6.718 \times 10^{16}$ m$^5$/s, which is almost the same as that after merging. Thus, by accounting for the orbital AM, the total AM is conserved. In the second case, the total AM before and after merging was $2.070\pm0.02 \times 10^{17}$ m$^5$/s and $2.448\pm0.022 \times 10^{17}$ m$^5$/s, respectively. The eddies co-rotated around the center of mass with an angular speed $\omega$ of $-1.44°$/day, which provided an orbital AM of $0.343 \times 10^{17}$ m$^5$/s. In addition, if the water above the two subsurface AEs was involved in such orbital AM due to the Taylor–Proudman effect,

there should be an additional amount of $0.110 \times 10^{17}$ m$^5$/s. According to the Taylor–Proudman theorem, the water was attached to the moving eddies under the geostrophic condition. Thus, the total orbital AM is $0.453 \times 10^{17}$ m$^5$/s, approximately 22% of the AM in both eddies. After accounting for the orbital AM to the system, the total AM is approximately $2.523 \times 10^{17}$ m$^5$/s, which is almost the same as the value after merging. In both cases, the orbital AM is non-negligible.

The law of conservation of total AM was used in previous theoretical models, but without consideration of orbital AM. These calculations of total AM assumed that eddies were immersed in a quiescent fluid [Pavia and Cushman-Roisin, 1990; Lumpkin et al., 2000], i.e., eddies were motionless before merging. However, this is not true because two approaching eddies would rotate around the center of mass according to fluid dynamics. This rotation was also noted in previously observed eddy merging events [Schultz Tokos et al., 1994; Li et al., 2016]. Because the previous theoretical studies overlooked orbital AM, the calculations led to an incorrect AM balance between pre- and post-merging stages.

Although the laws of conservation of total circulation and conservation of total AM are associated with rotation of water, they are somewhat different from the above analysis. The law of conservation of total circulation is only associated with eddy circulation, which is easy to use in applications. However, the conservation of total AM should consider both eddy circulation, referring to the center of each eddy, and orbital AM, referring to the whole system, which is difficult to calculate in complex environments.

In both cases, the total EKE decreased by a small amount after merging due to fusion. In the first case, the total EKE decreased from an initial range of 0.121 PJ to 0.094 PJ (0.021 m$^2$s$^{-2}$ to 0.016 m$^2$s$^{-2}$) before merging and then increased from 0.105 PJ to 0.116 PJ (0.017 m$^2$s$^{-2}$ to 0.020 m$^2$s$^{-2}$) after merging. The co-rotation also contributed approximately 3.4 TJ (5.8 $\times$ 10$^{-4}$ m$^2$s$^{-2}$), a negligible value for total EKE. In the second case, the total EKE decreased from an initial range of 0.692±0.01 PJ to 0.516±0.004 PJ (0.056 m$^2$s$^{-2}$ to 0.041 m$^2$s$^{-2}$) after merging. The co-rotation also contributed a negligible value of 0.0068 PJ (5.5 $\times$ 10$^{-4}$ m$^2$s$^{-2}$) in total EKE. Such a decrease in total EKE was noted in a previous study [Nof, 1990], in which EKE was the only component of energy in the theoretical one-layer model.

On the contrary, the total PE increased significantly after merging. In the first case, the total PE increased by approximately 0.207 PJ (0.035 m$^2$s$^{-2}$), from an initial range of 0.241±0.008 PJ to 0.448±0.01 PJ (0.041 m$^2$s$^{-2}$ to 0.076 m$^2$s$^{-2}$). In the second case, the total PE increased by approximately 1.343 PJ (0.096 m$^2$s$^{-2}$), from an initial range of 5.229±0.07 PJ to 6.572±0.04 PJ (0.423 m$^2$s$^{-2}$ to 0.519 m$^2$s$^{-2}$). The large increase of PE cannot be explained by the loss of EKE because eddy PE is, in general, an order of magnitude larger than the EKE [Su and Ingersoll, 2016]. Thus, additional PE from the background environment contributed to the EPE, supporting the merging events, and the total mechanical energy increased after merging. Both the summer and autumn seasons are favorable for this merging condition in oceans of the Northern Hemisphere, which consequently leads to fewer eddies in these seasons, as noted before. This leads to the question of where the huge additional PE came from.

To find the source of the EPE increase, we calculated the change of eddy gravitational PE, referring to the background sea level, with eddy shift parameter $b$, according to Eq. (10). In the first case (Fig. 3a), $b$ decreased from 0.045 m to 0.038 m after merging. This slight vertical sink of the eddy released approximately 0.401 PJ (0.069 m$^2$s$^{-2}$) of gravitational PE, 51.6% of

which accounts for the EPE increase. In the second case (Fig. 5a), $b$ decreased from 0.02 m to 0.001 m after merging. The vertical sink of the eddies released approximately 2.29 PJ (0.185 $m^2s^{-2}$) of gravitational PE, 58.6% of which accounts for the EPE increase. Thus, the vertical sink of the eddies released gravitational PE that supported the EPE increase. In addition, it can be deduced that the sink occurred at the level below the eddies; otherwise, it could not provide sufficient gravitational PE. For example, if only the surface water sank to the level of $b$, the released gravitational PE would be much less than the aforementioned values. This implies that such gravitational PE may be released from the ocean interior, deep below the upper ocean layer. It is also deduced that subsurface eddies might have larger PV than surface eddies. A deeper eddy may have larger gravitational PE. Thus, merged eddies have additional PE conversion from the background gravitational PE below the eddies, which has not been noted previously.

Similar to the results of some previous theoretical studies [Gill and Hopfinger, 1987; Pavia and Cushman-Roisin, 1990], the total mechanical energy increased after merging, and vice versa, in the present work. On the other hand, eddies will merge if external energy is input into them. An underrated paper illustrates such a phenomenon [Carnevale and Vallis, 1990]. Compared with these studies, our new findings are that the eddy PE dominates the increase of total mechanical energy (i.e., the sum of EKE and EPE) and that the EPE increase is converted from the eddy body sink.

Although the total circulation was conserved, the mean PV decreased after the merging in both cases due to the increase of eddy area. PV conservation has been generally assumed in theoretical models [Cushman-Roisin, 1986; Pavia and Cushman-Roisin, 1990; Nof, 1990], although others have stressed that the PV of eddies alternates during the interaction of merging [Gill and Hopfinger, 1987; Nof and Simon, 1987]. Here, we point out that the total circulation conservation other than the individual eddy PV conservation becomes a constraint for eddy merging.

The eddy enstrophy decreased after merging, becoming even smaller than the mean enstrophy of the eddies (Fig. 3c and Fig. 5c). Because the EKE dissipation rate is proportional to the eddy enstrophy [Li et al., 2018], the merged eddies have a smaller dissipation rate, which might support a potentially longer eddy lifetime.

Table 1. Observed physical quantity changes after merging events and conjectured changes after splitting events

| Physical Quantity | Surface Merging | Subsurface Merging | Eddy Splitting |
|---|---|---|---|
| Mass | Conserved | Conserved | Conserved |
| Circulation | Conserved | Conserved | Conserved |
| Angular momentum | Conserved | Conserved | Conserved |
| Potential vorticity | Decreased | Decreased | Increased |
| Enstrophy | Decreased | Decreased | Increased |
| Eddy kinetic energy | Decreased | Decreased | Increased |
| Eddy potential energy | Increased | Increased | Decreased |
| Mechanical energy | Increased | Increased | Decreased |

The observed changes of physical quantities after the merger events are listed in Table 1. Only three physical quantities (mass, circulation, and angular momentum) are conserved; the others have notable changes. The eddy merger is represented simply in the schematic diagram below (Fig. 6). Two faster and smaller eddies approach each other before the merger. During this time, they co-rotate anticyclonically with the water trapped by the eddies. Then, the two eddies coalesce into a slower and larger eddy. Both the surface eddy merger and the subsurface eddy merger satisfy the same conservation and conversion laws,

as addressed above. The only annotation is that the water above the two subsurface AEs should be involved due to the Taylor–Proudman effect.

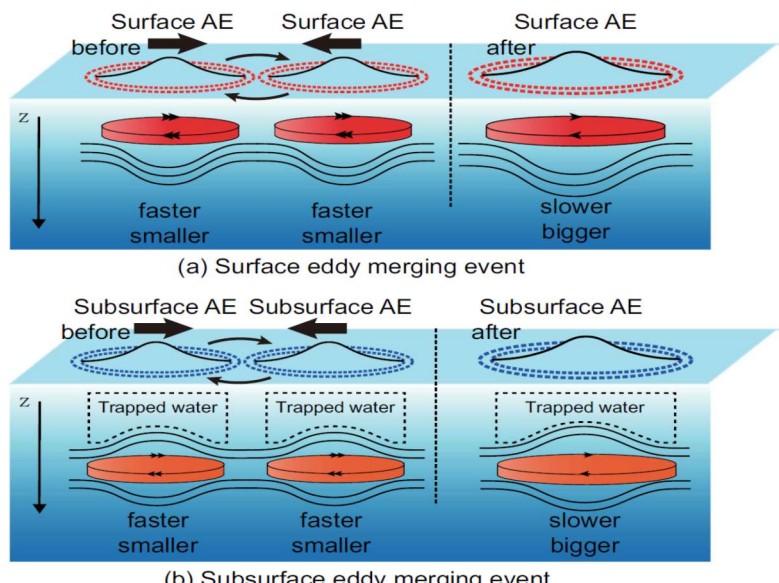

Figure 6. (a) Surface eddy merging event and (b) subsurface eddy merging event.

Eddy merging events could play a role in inverse energy cascades from small scale to large scale in two ways. On one hand, merging can pump EKE and EPE from small-scale eddies to large-scale eddies. On the other hand, the EKE of large-scale eddies has a lower dissipation rate and longer residence time. Thus, eddy merging functions likes a "large-scale energy pump" in the inverse energy cascades.

There is a long-term problem of which physical processes govern the seasonal variability of EKE [Marshall et al., 2002].

The eddy merging process provides an effective means of mesoscale genesis, which might be a link in the chain of events of such problem. In the Northern Hemisphere (e.g., the Gulf Stream region and the Kuroshio Extension region), the ocean is mostly baroclinically unstable in the mixed layer during late winter [Zhai et al., 2008; Qiu and Chen, 2010], when large-scale atmospheric forcing induces submesoscale eddies via mixed-layer instabilities [Sasaki et al., 2014]. The surface ocean then becomes strongly stratified due to surface heating and weak vertical mixing in summer [Zhai et al., 2008; Ma et al., 2014].

This strong stratification provides a large PE support for eddy mergers. Eddy merging events can pump the energy from

submesoscale eddies into mesoscale eddies, in which it persists due to weaker dissipation. Consequently, EKE peaks in summer, as observed in previous studies [Zhai et al., 2008; Sasaki et al., 2014]. Meanwhile, the submesoscale eddy condition itself usually has a seasonality, which affects the mesoscale eddy condition by inverse cascading [Yu et al., 2019]. The strong eddy activity in turn modulates the mixed layer depth [Gaube et al., 2019] and the isopycnals [Su et al., 2014].

## 4. Discussion

### 4.1 Sensitivity to parameters

Firstly, there may be some concern about the sensitivity of the results to the choice of the lateral boundary of integration. Because we used optimally fitted intrinsic parameters other than the originally identified boundaries (e.g., Fig. 2 and Fig. 4) for integration, the results are insensitive to the identified eddy boundaries. In addition, the SLA (and vorticity) contour lines are self-similar ellipses according to the eddy model in Eq. (1), so the lateral boundary of integration was chosen as an ellipse. The conservation relationships are insensitive to the choice of the ellipse. For example, if we extend the integration region to a larger self-similar ellipse with boundary satisfying $\xi = 0$, the integration properties of both sides (before merger and after merger) synchronously become larger with nearly the same ratio (Fig. 7).

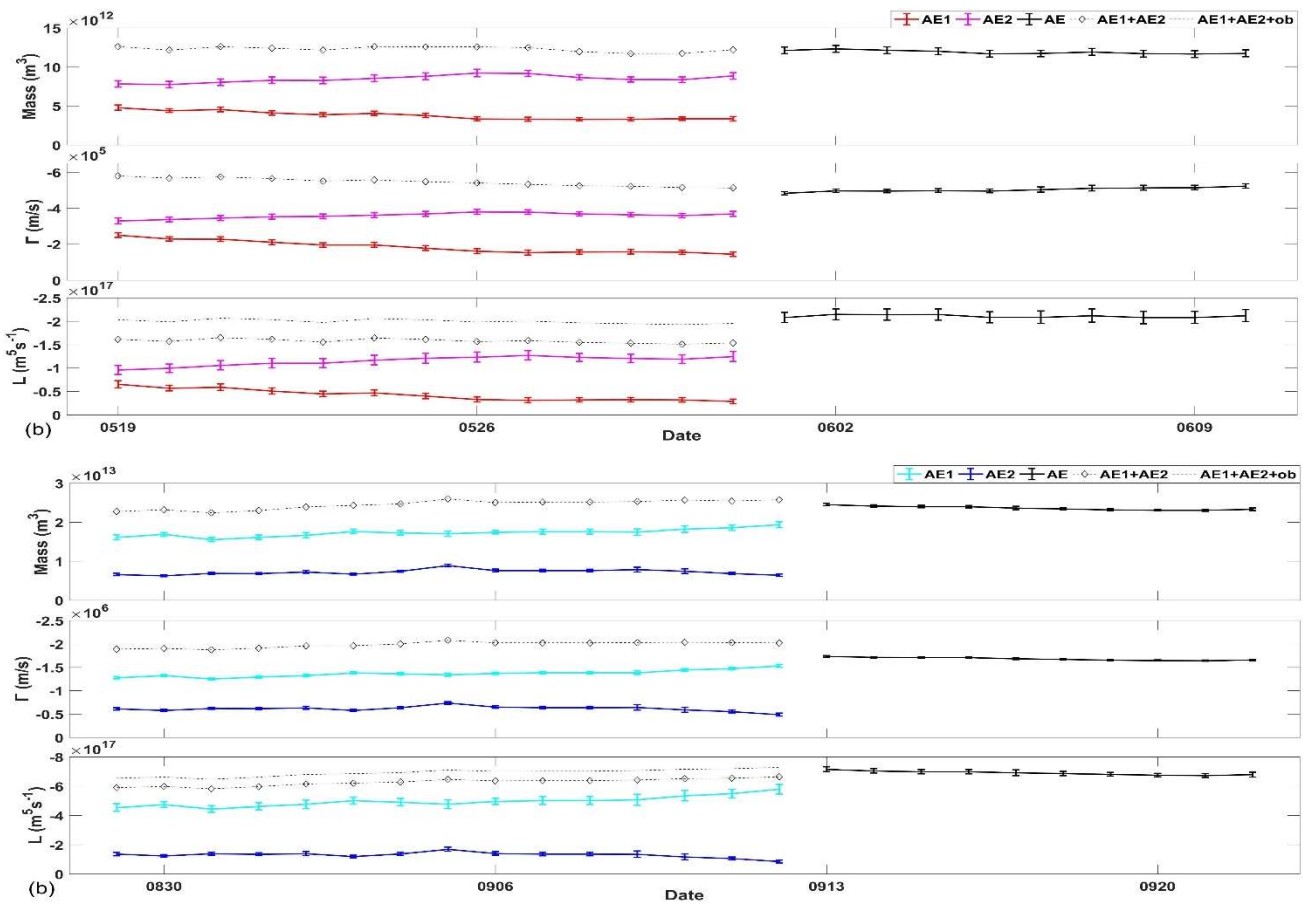

Figure 7. There are two panels. The eddy properties (mass, circulation, and angular momentum) with a larger lateral boundary of integration in the merging event of (a) surface mesoscale eddies and (b) subsurface mesoscale eddies.

   Secondly, there may also be a concern about how sensitive the results are to the choice of $H_1$, as $H_1$ was chosen as 200 m in the present study. Thus, we recalculated the eddy properties with the same eddy parameters obtained from the SLA field and

density profile according to GODAS data but simply enlarged $H_1$ from 200 m to 300 m while retaining the other parameters (including $h_1$ and $h_2$). The results are illustrated in Fig. 8. It is obvious that the total mass, total circulation, and total AM are hardly changed. Among these, the circulation seems more sensitive to the choice of $H_1$. This is true because the circulation depends on both the surface parameters and also the vertical parameters ($H_1$, $h_1$ and $h_2$) as the PV in Eq. (4). This might be the most sensitive case. In a more realistic scenario, a larger $H_1$ would decrease the density difference $\rho_2 - \rho_1$ between the lower

levels $H_1$ and $H_2$, which consequently would lead to a larger $h_2$. Thus, the PV in Eq. (4) should be less sensitive to $H_1$ due to the compensation of $h_2$.

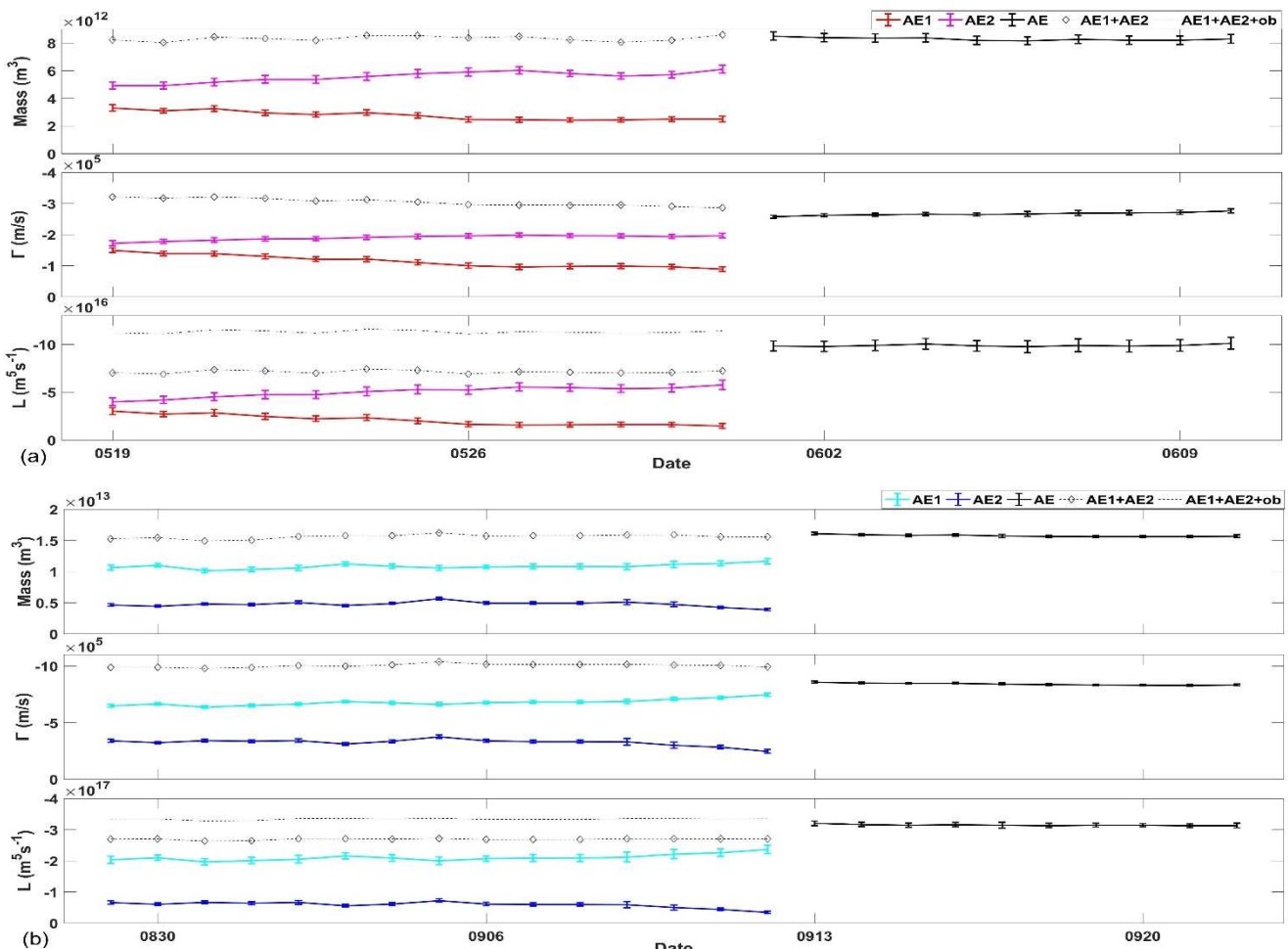

Figure 8. The eddy properties (mass, circulation, and angular momentum) with a thicker $H_1 = 300$ m in the merging event of (a) surface mesoscale eddies and (b) subsurface mesoscale eddies.

In contrast to previous simple models, there are six eddy parameters—three horizontal ($A$, $L_x$, and $L_y$) and three vertical ($H_1$, $h_1$, and $h_2$) in the present model. Two additional eddy parameters referring to the background—vertical shift $b$ and angular speed $\omega$, are also used. The additional degrees of freedom were expected to solve the dilemma of parameters being less numerous than conservation laws. All of the potential conservation laws were expected to hold simultaneously under the condition of more degrees of freedom than conservation laws. However, only limited conservation laws were found in the present results. It is noted that eddy conservation and conversion laws depend on the laws of physical dynamics, even if additional degrees of freedom can be provided in a mathematical model. This implies that the previous theoretical studies should be fully revisited because the physical conservation laws and physical properties were not correctly applied in these studies.

Although only eddy merging events were investigated in this study, it can be hypothesized that similar conservation and conversion laws are also valid in eddy splitting processes. The main difference may be that a physical quantity that ascends in the merging process, descends in the splitting process, and vice versa (Table 1). This hypothesis is partly supported by a previous numerical study. In a circulation-preserved flow field, a large eddy would split into small ones if energy decreased [Carnevale and Vallis, 1990]. Moreover, there are other causes (e.g., instability processes of the eddy itself and interaction with external flow) that may lead to eddy merging and splitting processes. In such processes, the conservation and conversion laws may be quite different from those in Table 1.

## 4.2 Vertical merger

In the section above, we examined the evolution of two subsurface anticyclones in 2015 during a merger event. As both AEs were subsurface eddies, one wonders whether it is a stacking process or whether the two cores coalesce. The stacking process is different from the two cores coalescing in two ways. The first is eddy area. In a stacking process, the area of the final eddy would be significantly smaller than the summed area of the two eddies. If two cores coalesce, the area of the final eddy would be nearly the sum of the two eddies because the water of both eddies coalesces. The second one is AM. In a stacking process, the AM of the final eddy would nearly be the sum of the two eddies. In the case of two cores coalescing, the AM of the final eddy would be significantly larger than the sum of two eddies due to orbital AM. The above eddy mergers satisfy such properties, so they are more like two cores coalescing than a stacking process. As both eddies were located in the same depth before their merger, the eddy merging events in this study should be considered as the horizontal merging type.

Additionally, there is a vertical merging type in which eddies are located in different depths before their merger. For example, Cresswell (1982) reported two warm-core eddies, Maria and Leo, located at 110–190 m and 280–470 m depth, respectively. Eddy Maria overlapped eddy Leo in a stacking-like process, and the two eddies finally coalesced. As there are so many surface

and subsurface eddies in the ocean, such merging types should also be common in nature. However, understanding how eddy properties satisfy the conservation and conversion laws after merger remains a challenge.

Finally, there are some interesting problems associated with the present study: How does a subsurface eddy move vertically, and which eddy properties are preserved during the motion? These problems have seldom been considered as only horizontal motions (and/or transports) of eddies were considered in the previous studies. Similar to its horizontal motion, the vertical motion of an eddy might contribute to vertical transports and vertical mixing. This hypothesis should be studied further in the future.

**5 Conclusion**

In this paper, the long-term theoretical "energy paradox" of whether the final state of two merging anticyclones contains more energy than the initial state was studied by observation of two cases of eddy merger. Several conservation laws were examined using a two-layer model with parameters fitted to observations, as listed in Table 1. However, only two conservation laws (mass and total circulation) held with the eddy parameters. The third conservation law for AM also held when it was accounted for properly by including the orbital AM, a previously overlooked value. Both circulation conservation laws and orbital AM were overlooked in previous theoretical studies. This study provides new insights beyond those of previous similar studies, including explaining some previous points of confusion.

In contrast, neither the EKE nor the EPE were conserved after merging. The EKE decreased due to fusion and the EPE increased due to environmental PE conversion related to the vertical shift parameter $b$. The total mechanical energy increased after merging. According to the present results, we can answer the energy paradox: the final merged eddy has more energy than the initial state eddies, and the energy is mainly contributed from background gravitational PE below the eddies, which is converted to EPE. In addition, eddy merging behaves like a "large-scale energy pump" in inverse energy cascades, which plays an important role in ocean dynamics.

Finally, the merging and splitting of eddies do not change the total mass, circulation, and AM of the flow field system, but they do change the energy distributions and portions in different scales, which is essential for the energy cascade in multiscale fluid dynamics.

*Author contributions*. L.S. designed the research; Z.F.W. and L.S. performed the research; Q.Y.L. and H.C. contributed data; Z.F.W. and H.C. drew figures; and L.S and Z.F.W. wrote the paper.

*Competing interests*. The authors declare no conflict of interest.

*Acknowledgements*. We thank the reviewers and the editor for their useful comments. This work was supported by the National Foundation of Natural Science of China (No. 41876013), the National Programme on Global Change and Air–Sea Interaction (GASI-IPOVAI-04). We thank AVISO for the SLA data (http://www. aviso.oceanobs.com/), APDRC/IPRC for the SST data

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
