# Peer review of "Two typical merging events of oceanic mesoscale anticyclonic eddies"

_Ocean Science, 2019_

## Referee Comment (RC1) · Anonymous Referee #1 · 10 Jul 2019

This study nicely examines the potential conservation and conversion laws during the merging of eddies, using observational data and theoretical models. This research leads to some new result that total mass (volume), total circulation (area integration of vorticity) and total angular momentum (AM) are conserved under given certain conditions, which provide new insights from previous similar studies including explaining some previous confusions. This study would be interesting to the research community of coherent eddy dynamics. I do have a few medium to minor comments that the authors need to consider carefully before it may be accepted.

* line 65: "the products are available on a daily scale with a $0.25° \times 0.25°$ resolution in the global ocean as DUACS DT14 [Pujol et al., 2016]."

It is important to remind the readers that this 0.25 degree resolution is only the data

resolution, not the physical signal resolution. The real signal resolution of Aviso is mostly only 100-200 km.

* line 100: "In the present study, both H 0 and H 1 are chosen to be 200 m, partly according to some recent observations"

is your result sensitive to your choice of 200m? Need some discussion here.

* line 20: "During their lifetime, complex dynamic processes occur, such as merging and splitting, which are associated with an eddy's genesis and termination. "

While eddy merging and splitting are an important topic, please clarify that you mainly focus on coherent eddies in this study (e.g. those you can count and recognize) rather than general eddy field. Note that eddies include not only coherent vortexes (your focus) but also all the rotational but incoherent turbulent structures such as chaotic filaments and fronts. Most of eddy kinetic energy (EKE) in the ocean are not from coherent eddies but from incoherent ones; and eddy transport of tracers is mostly due to incoherent motions: e.g. see and cite the following papers: Partitioning Ocean Motions Into Balanced Motions and Internal Gravity Waves: A Modeling Study in Anticipation of Future Space Missions, Journal of Geophysical Research, 123, 8084–8105 and this paper: Ocean submesoscales as a key component of the global heat budget. Nature Communications, 9, 775. Another example is your line 75 "Surface eddies are distinguished from subsurface eddies by whether their core is in the surface layer or located inside the water column (Fig. 1a)". Incoherent eddies usually do not have a core and do not have the concept of eddy radii. This is not a trivial comment and you should treat seriously: your first paragraph seems to mix/confuse these two together.

* line 245: "we calculated the change of eddy gravitational PE"

Most people will not understand this term. Define "eddy gravitational PE", its meaning and difference from EPE and indicate how you calculate it.

* around line 280: "This strong stratification provides a large PE support for eddy merg-
ers." Is this correct? usually a stronger stratification has a weaker PE, e.g. see QG PE density b'ˆ2/b_z

This is nice but it will benefit the readers by citing related papers here such as the paper on the nonlinear interaction of eddies (e.g. inverse cascade): e.g. a review paper Klein et al. 2019. Ocean-Scale Interactions from Space. Earth and Space Science, 6, 795-817.

* line 260: "eddy PE dominates the increase of total mechanical energy, and that the EPE increase is converted from the eddy body sink."

Most people will get lost by what you mean of "mechanical energy". Do you mean EKE + EPE? Please explain clearly. Also, explain what you mean by eddy body sink and why you have this sink? Avoid unusual jargon as much as possible.

* line 240: "The large increase of PE cannot be explained by the loss of EKE, since that eddy PE is, in general, an order of magnitude larger than the EKE"

This is correct but it is better to support this by citing related papers here such as this one: On the Minimum Potential Energy State and the eddy-size-constrained APE Density. JPO, 46, 2663–2674.

* This paper use the method of a two-layer model, which has its advantage but you should discuss the limitation caused by using this simple model. E.g., discuss how much uncertainty it may cause.

* line 274: "The eddy merging process provides an effective means of mesoscale genesis, which might be a link in the chain for another long-term problem of what physical processes govern the seasonal variability of EKE [Marshall et al., 2002]."

Eddy merging is indeed a potential important mechanism affecting eddy seasonality. But you should mention explicitly here that submesoscale itself usually has a seasonality (which affect mesoscale by inverse cascade). For example, recently there is a significant observation in North Atlantic about the seasonality of submesoscale, which

you may cite: Yu et al. 2019. An Annual Cycle of Submesoscale Vertical Flow and Restratification in the Upper Ocean. JPO, 49, 1439–1461.

+++++++++++++++++++ minor comments:

* line 201: "we find the second conservation law of total circulation. "

Do you mean "we find that the second conservation law of total circulation holds"? Why call it second conservation law? do you invent this term? Do you mean the second conservation law is about the conservation of total circulation? It reads confusing.

* around line 25: please specify the structures/sections of your paper here.

* around line 90: "For a two-layer model, . . ." Do you mean you use a two-layer model? or this is is set up of a usual two-layer model?

* line 120: "The first merging event . . ." what do you mean by "first" here? relative to what?

* around line 140: "It is noted that the vorticity of AE2 is significantly smaller, although it had a larger amplitude."

what quantity do you mean here for larger amplitude? It is confusing.

* line 192: "Finally, we calculated the energies of eddies. Both the EKE and EPE had similar variations before merging."

So what? any explanation or implication by this result? clarify what is the point here?

* line 230: "which is hardly calculated in complex environments." Do you mean "which is hard to calculate" here?

* around line 280: "The strong eddy activity in turn modulates the mixed layer depth [Gaube et al., 2019]."

This is correct but it is very helpful to mention that eddy activity in general modulate the isopycnals (more than just mixed layer depth), e.g. may see and cite this paper: An

idealized model of Weddell Gyre export variability. JPO, 44, 1671-1688.

* around line 255: "A rarely known paper illustrates such a phenomenon [Carnevale and Valli's, 1990]."

The sentence is awkward; suggest to remove the word "rarely known".

* line 201: "In both cases, the total circulation of the eddies seldom changes."

Please specify number or figure to show this result, if any

* line 266: "The eddy enstrophy also decreased after merging, even smaller than mean enstrophy of eddies."

Specify the figures for this result, if any.

* line 232: "0.121 PJ to 0.094 PJ" The unit of PJ is awkward here; no one will have a feel on it. Please change to (m/s)ˆ2

---

## Referee Comment (RC2) · Anonymous Referee #2 · 16 Jul 2019

General comments :

This manuscript applied conservation laws to two merger events observed by satellite in the South China Sea. Sea-level anomaly and temperature are used to determine the eddy type (surface or subsurface). Two cases are then detailed : first, the merging of two surface anticyclones, then of two subsurface ones. Vertical ocean structure is given by GODAS Ocean re-analysis and a 2-layers model is fitted to observations. Evolution of the eddies parameters, circulation, angular momentum, vorticity, energy and enstrophy are observed during the merging processes. The main findings are that angular momentum is conserved (unlike PV) when taking into account for the orbital momentum of the pair of eddies rotating around each other, and a significant decrease of EPE is observed and interpreted as due to the sinking of the eddies during merger.

[Figure]

I found the manuscript generally well organized, but still needs to be clarified. The language requires some correction too. Figures are useful and illustrative. I believe this manuscript represents an important contribution to the knowledge of mesoscale eddy mergers, but in order to be accepted for publication in Ocean Science, I recommend a major revision to be done following the comments listed below.

Specific comments :

- About the method : I couldn't find how h1 and h2 were determined in the text. How the interface of the eddies were chosen according to its vertical structure? Data and Method section needs to clarify this point, and also readers would appreciate to see the vertical structure of the eddies/background as an illustration and visual check.

- It is not clear how eddy properties are considered below the surface. If I understand correctly what is here done, eddy properties inferred from surface observations are taken as average over the layer defining the eddy. If this can be acceptable for surface anticyclones, I wonder if this assumption not too strong when considering subsurface eddies with subsurface velocity maximum? What is the depth a typical subsurface eddies in the South China Sea? Alternatively, velocity fields from ocean re-analysis can be used in the considered layer.

- H1 is chosen as a constant value, but in the real ocean, this likely not true and can lead to substantial variation in eddy properties. The reference provided to justify this choice are from different places with different stratification. How sensitive are the result to the choice of H1?

- How are the lateral boundary of integration chosen? This is not details neither, and, I presume, can lead to significantly different results. Again, how sensitive are the results to the choice of this parameter?

- The paper lacks of a statistical generalization of the results. Have you studied other examples of merging before choosing to focus on the two presented in the paper? I

presume that once the work is achieved for two examples, it can easily be applied on others examples. Otherwise, based on only two examples, the conclusions about generalization of the conservation rules, and splitting, needs to be mitigated.

Technical corrections :

l19 "by trapping them", please rephrase l20 "the most energetic component in the ocean", please provide reference l22 "eddy's life-cycle and transports." l24 "than before", please be more precise (than pre-existant eddies, than the sum of the two original eddies...) l25 "by Gill and Griffiths", Is there no reference for this work? l28 "Pandora's box", l40 & 300 "are less than", please specify ("less numerous") and correct in the whole manuscript l45 "field", please prefer research cruises to "voyage" l60 "eddy merging", please rephrase "after two typical eddy mergers", merging is not a noun... l65 "for the global" l80 "as previously used", please provide reference l89 "eddy area but eddy radii is an extensive quantity", please rephrase this is not clear l92 "compositing", this is not a verb, please rephrase l96 Do you mean " too small and can be ignored"? l106 PV anomaly? Please provide a reference or a demonstration that the average circulation is equal to the surface integrated PV anomaly. l110 Where is the x- and y-axis origin? l112 u and v refer to surface eddy velocity but considered as average swirling velocity of the eddy, right? l114 & l116 Why is there no (H1+h1+h2) factor in the integral? l114 Please provide the expression of the reduced gravity. l116 Please provide a definition and expression for ksi. Again this is for a surface parameter I presume. l126 Please provide number in meter too for h2. l127 "the parameters of both eddies" l135 & 184 "experienced changes" l158 "described in the previous" l161 Please provide number in meter for h1 and h2. l156-161 Consider moving some of this part to methods with more detailed explanation on the choice of the density interface rho0, rho1 and rho2. l165 "came close to each other with a" l166 "this subsurface merging event" l175-177 Is it a stacking process? or the two cores coalesce? How the vertical structure of the eddies evolve during the merger? l185 This is wrong, now h1 is same order as h2. l198 and following : What the +/- corresponds to? Please be consistent with

number of significant digits between parameters (sometimes 3, sometimes 4, I would give 2). l203 Please provide a reference l209 "as mentioned previously" l219 "merging" "non-negligible" l241 "sported"? l242 "in the northern ocean"? l245-247 How is the eddy gravitational PE wrt background sea level computed? Please provide the formula applied here to infer the numbers. l258 "rarely-evoked" "poorly-known" "underrated" l264-265 Please rephrase this sentence is not clear. l266 "enstrophy decreased" l272 "in the inverse energy cascade" l275 "mostly baroclinically" l280 "persists" l285 "observation of two cases of eddy merger" l286 "fitted to" l289-290 " Thus, parts of these ... in future." What do you mean here? l297 H1 is fixed here... l308-309 Why eddy splitting will work similarly than merging? Splitting can have very different causes (instability processes of the eddy itself, or interaction with external flow) and might not work the same way as mergers work. The authors should prove or illustrate their statement with an example, or remove the last column of table 1 and mitigate their conclusions.

Figure 1 : The top panels are suited to introduce the 2-layer model, while the bottom panels already detail some results. Please split into two figure with one put at the end of the manuscript with the conclusion. In (b) please draw isopycnals as lines, the colors are confusing. Please also specify H0 and H2. Figure 2 and 4 : Please mark the eddies described in the text (AE1, AE2, A1, A2 and A) Figure 3 and 5 : Please give more details, what is "A2+A2+ob" for instance? Hard to know without reading carefully the paper.
* * *

---

## Editor Comment (EC1) · Ilker Fer (Editor) · 18 Jul 2019

Dear Zi-Fei,

I will request that you improve the analysis and presentation (e.g. description of calculations for some parameters is not clear, errors are not discussed). In particular, you need an estimate of errors associated with the analysis, following from the observations and methods. Figures 3 and 5 must have errorbars (at least on relevant parameters where possible). Looking at Fig 2, and definition of eddy boundaries, there seems to be an arbitrariness and associated uncertainty. Also please consider the following details.

1. Please see if you can have the language and style improved with help from colleagues. Alternatively you can consider professional services. Also please check for

typos throughout.

2. Make sure all cites are included in the reference list, e.g. Gill and Griffiths

3. I like one referee's suggestion of splitting Fig 1 into two and showing the lower panels later toward the conclusions.

4. li45-46: There has been some observational work from the Lofoten Basin (I recall Roshin Raj's paper demonstrating some mergers).

5. li54: "without any assumption" is a *very* strong statement. You do use 2-layers (or 1 or 3), approximate SLAs as Gaussian, assume H0=H1=200m, and calculate the velocity and vorticity from geostrophy! etc..

6. li76: SST is not the only contributor to density. Is this a regional statement?

7. Eq.2 and on with integrals: are these accurate? It's not clear to me how you define the volume. How are the anomalies of u and rho calculated? Please describe how you obtained h2.

8. li198-199: now we have plus/minus (which is good), but what are these? Standard error, standard deviation, uncertainty? Please describe.

Thank you, Ilker

───────────────────────────

---

## Author Comment (AC1) · 27 Aug 2019

We thank you for the insightful comments and useful suggestions.

Q: 1. Please see if you can have the language and style improved with help from colleagues. Alternatively you can consider professional services. Also please check for typos throughout.

A: We have applied professional services for language.

Q: 2. Make sure all cites are included in the reference list, e.g. Gill and Griffiths

A: We have added the references.

Q: 3. I like one referee's suggestion of splitting Fig 1 into two and showing the lower

panels later toward the conclusions.

A: We have split it into two by following your and reviewer's suggestion.

Q:4. li45-46: There has been some observational work from the Lofoten Basin (I recall Roshin Raj's paper demonstrating some mergers).

A: We have cited a work from the Lofoten Basin (Bashmachnikov et al., 2017). Now we add Raj's paper (Raj et al., 2016).

Q:5. li54: "without any assumption" is a *very* strong statement. You do use 2-layers (or 1 or 3), approximate SLAs as Gaussian, assume $H0=H1=200m$, and calculate the velocity and vorticity from geostrophy! etc.

A: We are sorry for the unclear statement. The assumption is not for calculation itself, but for conservation law. We have modified it.

Q:6. li76: SST is not the only contributor to density. Is this a regional statement?

A: Yes, it is a regional statement. Although both SST and SSS contributes to density, density anomaly is dominated by SST anomaly since SSS anomaly is very small.

Q:7. Eq.2 and on with integrals: are these accurate? It's not clear to me how you define the volume. How are the anomalies of u and rho calculated? Please describe how you obtained h2.

A: Yes. We have added these accordingly following your and reviewers' suggestions.

Q:8. li198-199: now we have plus/minus (which is good), but what are these? Standard error, standard deviation, uncertainty? Please describe.

A: They are the standard deviation. We add these in text.

---

## Author Comment (AC2) · 31 Aug 2019

We thank Referee #1 for the useful comments and suggestions.

The main revisions includes:

1. All figures are redrawn accordingly.
2. We clarify the data and method with more details and formulas.
3. The sensitivity of result to parameters are discussed in a new subsection.
4. The possible vertical process are also discussed in a new subsection.
5. The changes according to Referee #1, #2, and editor are marked with red, blue, and green, respectively.

Q:* line 65: "the products are available on a daily scale with a 0.25_ _ 0.25_ resolution in the global ocean as DUACS DT14 [Pujol et al., 2016]."
It is important to remind the readers that this 0.25 degree resolution is only the data resolution, not the physical signal resolution. The real signal resolution of Aviso is mostly only 100-200 km.

A: Thanks, we have added this notation accordingly.

Q:* line 100: "In the present study, both H 0 and H 1 are chosen to be 200 m, partly according to some recent observations" is your result sensitive to your choice of 200m? Need some discussion here.

A: We have added the discussion in a new section accordingly.

Q:* line 20: "During their lifetime, complex dynamic processes occur, such as merging and splitting, which are associated with an eddy's genesis and termination. " While eddy merging and splitting are an important topic, please clarify that you mainly focus on coherent eddies in this study (e.g. those you can count and recognize) rather than general eddy field. Note that eddies include not only coherent vortexes (your focus) but also all the rotational but incoherent turbulent structures such as chaotic filaments and fronts. Most of eddy kinetic energy (EKE) in the ocean are not from coherent eddies but from incoherent ones; and eddy transport of tracers is mostly due to incoherent motions: e.g. see and cite the following papers: Partitioning Ocean Motions Into Balanced Motions and Internal Gravity Waves: A Modeling Study in Anticipation of Future Space Missions, Journal of Geophysical Research, 123, 8084–8105 and this paper: Ocean submesoscales as a key component of the global heat budget. Nature Communications, 9, 775. Another example is your line 75 "Surface eddies are distinguished from subsurface eddies by whether their core is in the surface layer or located inside the water column (Fig. 1a)". Incoherent eddies usually do not have a core and do not have the concept of eddy radii. This is not a trivial comment and you should treat seriously: your first paragraph seems to mix/confuse these two together.

A: Thanks for the useful information, we have clarified this according to your suggestion. We also add "Besides, there are incoherent eddies, which usually do not have a core and do not have the concept of eddy radii. These incoherent eddies are also important, since most of eddy kinetic energy (EKE) in the ocean are from incoherent ones [Torres et al., 2018]; and eddy transport of tracers is mostly due to incoherent motions [Su et al., 2018]".

Q:* line 245: "we calculated the change of eddy gravitational PE" Most people will not understand this term. Define "eddy gravitational PE", its meaning and difference from EPE and indicate how you calculate it.

A: Suggestion followed, we have added the formula as Eq. (10).

Q:* around line 280: "This strong stratification provides a large PE support for eddy mergers." Is this correct? usually a stronger stratification has a weaker PE, e.g. see QG PE density b'^2/b_z This is nice but it will benefit the readers by citing related papers here such as the paper on the nonlinear interaction of eddies (e.g. inverse cascade): e.g. a review paper Klein et al. 2019. Ocean-Scale Interactions from Space. Earth and Space Science, 6, 795-817.
A: Suggestion followed, we have cited the paper.

Q:* line 260: "eddy PE dominates the increase of total mechanical energy, and that the EPE increase is converted from the eddy body sink." Most people will get lost by what you mean of "mechanical energy". Do you mean EKE+ EPE? Please explain clearly. Also, explain what you mean by eddy body sink and why you have this sink? Avoid unusual jargon as much as possible.
A: Yes, it is EKE+EPE. We have added Eq. (10) to illustrate this.

Q:* line 240: "The large increase of PE cannot be explained by the loss of EKE, since that eddy PE is, in general, an order of magnitude larger than the EKE" This is correct but it is better to support this by citing related papers here such as this one: On the Minimum Potential Energy State and the eddy-size-constrained APEDensity. JPO, 46, 2663–2674.
A: Thanks, we have added the reference.

Q:* This paper use the method of a two-layer model, which has its advantage but you should discuss the limitation caused by using this simple model. E.g., discuss how much uncertainty it may cause.
A: Thanks for the suggestion, we have added a new section to discuss this.

Q: * line 274: "The eddy merging process provides an effective means of mesoscale genesis, which might be a link in the chain for another long-term problem of what physical processes govern the seasonal variability of EKE [Marshall et al., 2002]." Eddy merging is indeed a potential important mechanism affecting eddy seasonality. But you should mention explicitly here that submesoscale itself usually has a seasonality (which affect mesoscale by inverse cascade). For example, recently there is a significant observation in North Atlantic about the seasonality of submesoscale, which you may cite: Yu et al. 2019. An Annual Cycle of Submesoscale Vertical Flow and Restratification in the Upper Ocean. JPO, 49, 1439–1461.
A: Thanks, we mention this explicitly according to your suggestion.

+++++++++++++++++ minor comments:
Q: * line 201: "we find the second conservation law of total circulation. " Do you mean "we find that the second conservation law of total circulation holds"? Why call it second conservation law? do you invent this term? Do you mean the second conservation law is about the conservation of total circulation? It reads confusing.
A: We are sorry for the unclear. The second conservation law is about the conservation of total circulation. We have modified it.

Q:* around line 25: please specify the structures/sections of your paper here.
A: We specify the structures/sections of the paper at the last paragraph of section 1.

Q:* around line 90: "For a two-layer model, : : :" Do you mean you use a two-layer model? or this is set up of a usual two-layer model?
A: A usual two-layer model.

Q:* line 120: "The first merging event : : :" what do you mean by "first" here? relative to

what?

A: we remove "first".

Q:* around line 140: "It is noted that the vorticity of AE2 is significantly smaller, although it had a larger amplitude." what quantity do you mean here for larger amplitude? It is confusing.

A: we are sorry for the confusing, it is eddy amplitude, a parameter associated with SLA in Eq. (1). We have clarified this.

Q:* line 192: "Finally, we calculated the energies of eddies. Both the EKE and EPE had similar variations before merging." So what? any explanation or implication by this result? clarify what is the point here?

Q:* line 230: "which is hardly calculated in complex environments." Do you mean "which is hard to calculate" here?

A: Yes. We have modified it.

Q:* around line 280: "The strong eddy activity in turn modulates the mixed layer depth [Gaube et al., 2019]." This is correct but it is very helpful to mention that eddy activity in general modulate the isopycnals (more than just mixed layer depth), e.g. may see and cite this paper: An idealized model of Weddell Gyre export variability. JPO, 44, 1671-1688.

A: Thanks for suggestion, we have added the words.

Q: * around line 255: "A rarely known paper illustrates such a phenomenon [Carnevale and Valli's, 1990]." The sentence is awkward; suggest to remove the word "rarely known".

A: Thanks for suggestion, "rarely known" is removed.

Q: * line 201: "In both cases, the total circulation of the eddies seldom changes." Please specify number or figure to show this result, if any

A: We add figures.

Q: * line 266: "The eddy enstrophy also decreased after merging, even smaller than mean enstrophy of eddies."Specify the figures for this result, if any.

A: We add figures.

Q: * line 232: "0.121 PJ to 0.094 PJ" The unit of PJ is awkward here; no one will have a feel on it. Please change to $(m/s)^2$

A: We have modified it accordingly.

---

## Author Comment (AC3) · 31 Aug 2019

We thank Referee #2 for the useful comments and suggestions.

The main revisions includes:

1. All figures are redrawn accordingly.
2. We clarify the data and method with more details and formulas.
3. The sensitivity of result to parameters are discussed in a new subsection.
4. The possible vertical process are also discussed in a new subsection.
5. The changes according to Referee #1, #2, and editor are marked with red, blue, and green, respectively.

Specific comments :

Q:- About the method : I couldn't find how h1 and h2 were determined in the text. How the interface of the eddies were chosen according to its vertical structure? Data and Method section needs to clarify this point, and also readers would appreciate to see the vertical structure of the eddies/background as an illustration and visual check.

A: We add the detail in section 2.3. The upper surface $h_1 = \frac{\rho_1}{\rho_1 - \rho_0} A$ and the lower surface

$h_2 = \frac{\rho_1}{\rho_2 - \rho_1} A$ satisfy $h_1 \sim h_2 \ll H_1$.

Q:- It is not clear how eddy properties are considered below the surface. If I understand correctly what is here done, eddy properties inferred from surface observations are taken as average over the layer defining the eddy. If this can be acceptable for surface anticyclones, I wonder if this assumption not too strong when considering subsurface eddies with subsurface velocity maximum? What is the depth a typical subsurface eddies in the South China Sea? Alternatively, velocity fields from ocean re-analysis can be used in the considered layer.
A: Yes, eddy properties inferred from surface observations are taken as average over the layer defining the eddy. The eddies are not in the South China Sea, but in the western tropical Pacific. Such thick of subsurface eddies are from previous observation [Li et al., 2017] and numerical simulation [e.g. Wang 2017].
Q:- H1 is chosen as a constant value, but in the real ocean, this likely not true and can lead to substantial variation in eddy properties. The reference provided to justify this choice are from different places with different stratification. How sensitive are the result to the choice of H1?
A: The sensitivity of result to H1 is discussed in a newly appended section 4.1.
Q:- How are the lateral boundary of integration chosen? This is not details neither, and, I presume, can lead to significantly different results. Again, how sensitive are the results to the choice of this parameter?
A: The sensitivity of result to lateral boundary is discussed in a newly appended section 4.1.
Q:- The paper lacks of a statistical generalization of the results. Have you studied other examples of merging before choosing to focus on the two presented in the paper? I presume that once the work is achieved for two examples, it can easily be applied on others examples. Otherwise, based on only two examples, the conclusions about generalization of the conservation rules, and splitting, needs to be mitigated.
A: Thanks for your comments, we have added the discussion of results in this new version.

Technical corrections :

Q:l19 "by trapping them", please rephrase

A: "by trapping those tracers along with the water"

Q:l20 "the most energetic component in the ocean", please provide reference

A:Thanks, reference added.

Q:l22 "eddy's life-cycle and transports."

A:Thanks, suggestion followed.

Q:l24 "than before", please be more precise (than pre-existant eddies, than the sum of the two original eddies...)

A:Thanks, "than the sum of the two original eddies".

Q: l25 "by Gill and Griffiths", Is there no reference for this work?

A: The reference is "GILL, A. E. and GRIFFITHS, R. W. 1981 Why should two anticyclonic eddies merge? In Ocean Modelling, 41. Unpublished manuscript."

Q:l28"Pandora's box",

A:Thanks, suggestion followed.

Q:l40 & 300 "are less than", please specify ("less numerous") and correct in the whole manuscript

A:Thanks, suggestion followed.

Q:l45 "field", please prefer research cruises to "voyage"

A:Thanks, suggestion followed.

Q:l60 "eddy merging", please rephrase "after two typical eddy mergers", merging is not a noun...

A:Thanks, suggestion followed.

Q:l65 "for the global"

A:Thanks, suggestion followed.

Q:l80 "as previously used", please provide reference

A:Thanks, suggestion followed.

Q:l89 "eddy area but eddy radii is an extensive quantity", please rephrase this is not clear

A:Thanks, suggestion followed.

Q:l92 "compositing", this is not a verb, please rephrase

A:Thanks, suggestion followed.

Q:l96 Do you mean " too small and can be ignored"?

A:Thanks, suggestion followed.

Q:l106 PV anomaly? Please provide a reference or a demonstration that the average circulation is equal to the surface integrated PV anomaly.

A: $\xi - f \frac{h_1 + h_2}{H_1}$ is PV anomaly (Gill A.E., p 192). Then

$$\Gamma = \iint \left( \xi - f \frac{h_1 + h_2}{H_1} \right) dx dy = \iint \left( \frac{f + \xi}{H_1 + h_1 + h_2} - \frac{f}{H_1} \right)(H_1 + h_1 + h_2) dx dy$$

Q:l110 Where is the x- and y-axis origin?

A: the x- and y-axis origin at eddy center.

Q:l112 u and v refer to surface eddy velocity but considered as average swirling velocity of the eddy, right?

A: Yes.

Q:l116 Why is there no (H1+h1+h2) factor in the integral?

A: We assume that only h1 and h2 change during the merging process but H1 does not change during the merging process. So only the potential energy associated with interface is considered [e.g., Lumpkin et al., 2000].

Q:l114 Please provide the expression of the reduced gravity.

A: suggestion followed.

Q: l116 Please provide a definition and expression for ksi. Again this is for a surface parameter I presume.

A: Yes, it is a surface parameter. And suggestion followed in Eq. (2).

Q:l126 Please provide number in meter too for h2.

A: suggestion followed.

Q:l127 "the parameters of both eddies"

A:Thanks, suggestion followed.

Q:l135 & 184 "experienced changes"

A:Thanks, suggestion followed.

Q:l158 "described in the previous"

A:Thanks, suggestion followed.

Q:l161 Please provide number in meter for h1 and h2.

A:Thanks, suggestion followed.

Q:l156-161 Consider moving some of this part to methods with more detailed explanation on the choice of the density interface rho0, rho1 and rho2.

A:Thanks, we have added more details in method.

Q:l165 "came close to each other with a"

A:Thanks, suggestion followed.

Q:l166 "this subsurface merging event"

A:Thanks, suggestion followed.

Q:l175-177 Is it a stacking process? or the two cores coalesce? How the vertical structure of the eddies evolve during the merger?

A: It is two cores coalesce. The stacking process is quite different from the two cores coalesce in that the area of "merged eddy" in stacking process is significantly smaller than the total area of two eddies. We did find examples of such stacking process.

Q:l185 This is wrong, now h1 is same order as h2.

A:Thanks, suggestion followed.

Q: l198 and following : What the +/- corresponds to? Please be consistent with number of significant digits between parameters (sometimes 3, sometimes 4, I would give 2).

A: Thanks, we have modified them.

Q: l203 Please provide a reference

A:Thanks, suggestion followed.

Q:l209 "as mentioned previously"

A:Thanks, suggestion followed.

Q:l219 "merging" "non-negligible"

A:Thanks, suggestion followed.

Q:l241 "sported"?

A:Thanks, suggestion followed.

Q:l242 "in the northern ocean"?

A:Thanks, "in oceans of the northern hemisphere. ".

Q:l245-247 How is the eddy gravitational PE background sea level computed? Please provide the formula applied here to infer the numbers.

A:Thanks, we add the formula in Eq. (10).

Q:l258 "rarely-evoked" "poorly-known" "underrated"

A:Thanks, suggestion followed.

Q:l264-265 Please rephrase this sentence is not clear.

A:Thanks, suggestion followed.

Q:l266 "enstrophy decreased"

A:Thanks, suggestion followed.

Q:l272 "in the inverse energy cascade"

A:Thanks, suggestion followed.

Q:l275 "mostly baroclinically"

A:Thanks, suggestion followed.

Q:l280 "persists"

A:Thanks, suggestion followed.

Q:l285 "observation of two cases of eddy merger"

A:Thanks, suggestion followed.

Q:l286 "fitted to"

A:Thanks, suggestion followed.

Q:l289-290 " Thus, parts of these ... in future." What do you mean here?

A: Removed.

Q:l297 H1 is fixed here...

A: Yes, modified.

Q:l308-309 Why eddy splitting will work similarly than merging? Splitting can have very different causes (instability processes of the eddy itself, or interaction with external flow) and might not work the same way as mergers work. The authors should prove or illustrate their statement with an example, or remove the last column of table 1 and mitigate their conclusions.

A: Thank for your comments. We have modified by following your suggestion.

Q: Figure 1 : The top panels are suited to introduce the 2-layer model, while the bottom panels already detail some results. Please split into two figure with one put at the end of the manuscript with the conclusion. In (b) please draw isopycnals as lines, the colors are confusing. Please also specify H0 and H2.

A: Suggestion followed.

Q: Figure 2 and 4 : Please mark the eddies described in the text (AE1, AE2, A1, A2 and A)

A: Suggestion followed.

Q: Figure 3 and 5 : Please give more details, what is "A2+A2+ob" for instance? Hard to know without reading carefully the paper.

A: Suggestion followed.

---

## Editor Comment (EC2) · Ilker Fer (Editor) · 7 Sep 2019

Thank you for your response to the reviewers' comments and the annotated version of your final response.

Before I can consider your submission in Ocean Science and send over for a re-review, I need you to address the following.

I find the introduction and use of errorbars unsatisfactory. It is mentioned only once in the figure caption, li 67: "errorbars indicate the standard deviation...", without further description. Because the bars are constant for a given parameter, I am guessing you obtained this as one std of say 10-15 values shown in the panels. Of course, for the relatively flat curves of AE, the std is minimal. But this doesn't mean that error

is so small. This is not an acceptable approach and I request you make an attempt to estimate an error based on your methods and analysis (and briefly describe this in the paper). If, in the light of errorbars, you need to adjust your discussion and/or conclusions, please do so.

Some minor edits: li 81: Eddy merger

li 86: the surface

li 87-88: please rewrite the opposite cases of surface and subsurface avoiding brackets

Table 1: Use increased/decreased (not ascended/descended). It would be better if you could add the percentage of increase/decrease too.

Fig 6 is horizontally distorted, please fix

Thank you, Ilker

---

## Author Comment (AC4) · 12 Sep 2019

Q: I find the introduction and use of errorbars unsatisfactory. It is mentioned only once in the figure caption, li 67: "errorbars indicate the standard deviation...", without further description. Because the bars are constant for a given parameter, I am guessing you obtained this as one std of say 10-15 values shown in the panels. Of course, for the relatively flat curves of AE, the std is minimal. But this doesn't mean that error is so small. This is not an acceptable approach and I request you make an attempt to estimate an error based on your methods and analysis (and briefly describe this in the paper). If, in the light of errorbars, you need to adjust your discussion and/or conclusions, please do so.

A: Thanks for comments and suggestions. Yes, we used the standard deviations of the values shown in the panels other than the standard deviations of values in each case. Now we estimate the errors based on each case.Now we estimate the errors based on each case, and add a new subsection 2.5 to descript this.

We first estimate the errors of eddy parameters (e.g., A, Lx, and Ly) obtained by non-linear fitting. This is simple, because the outputs of the fitting algorithm have already included the standard deviation (e.g., $\delta A$, $\delta b$ and $\delta S$) of each parameter, and the co-efficient of determination (R2). Typically, the standard deviations are 2$\sim$8% of eddy parameters, and the fitting performance R2 is from 0.87 to 0.98 in this study.

Secondly, we estimate the standard deviations of eddy properties. Since we have used numerical integration of eddy parameters to obtain eddy properties, there are no simple and explicit relations between eddy properties and eddy parameters. The exact standard deviations of eddy properties can hardly be obtained in this way. Here we approximately estimate the standard deviations of eddy properties by assuming that eddy is a circle with same area S of original ellipse. Then the eddy properties can be expressed as functions of eddy parameters (e.g. A and S) after integration. The standard deviations of eddy properties can now be estimated with standard deviations of eddy parameters. For example, the eddy enstrophy in Eq. (9) is Es=cA2/S, where c is the integration constant. Then the standard deviations of eddy enstrophy is $\delta Es=$ Es(2$\delta A$/A+$\delta S$/S). We use these standard deviations to draw errorbars in figures.

Some minor edits:

Q: li 81: Eddy merger

A:Thanks, suggestion followed.

Q: li 86: the surface

A:Thanks, suggestion followed.

Q: li 87-88: please rewrite the opposite cases of surface and subsurface avoiding

brackets

A:Thanks, suggestion followed.

Q: Table 1: Use increased/decreased (not ascended/descended).

A:Thanks, suggestion followed.

Q: It would be better if you could add the percentage of increase/decrease too.

A: Thanks for the suggestion. However, this might be not very significant, since the percentage of increase/decrease is not a constant. The previous theoretic studies have shown that the percentage of increase/decrease may vary in a wide range [e.g. Lumpkin et al., 2000; Sangra et al., 2005]. It depends on the parameters (size, amplitude, etc) of merging eddies before merger.

Q: Fig 6 is horizontally distorted, please fix

A:Thanks, suggestion followed.

---

## Author Response (AR3)

Q: P1 L10 Both conservation laws of circulation

A: suggestion followed.

Q: P1L20 and transport heat ... over long distances

A: suggestion followed.

You may also cite a recently accepted paper showing lagrangian trapping of floats for >1yr (https://www.nature.com/articles/s41598-019-49599-8)

A: suggestion followed.

Q: P1L25 because they contain most of .... and are responsible for most of the eddy transport...

A: suggestion followed.

Q:P2L48 you didnt take into account my remark about "voyages"

A: we have now modified voyages to research cruise.

Q:P4L88 please reformulate. "identify surface AEs (SSTA>0) and subsurface (SSTA<0)

A: suggestion followed.

Q:P6 L144 please refer to the method used (Wang et al?) to infer the coefficients

A: suggestion followed.

Q:P15 L334 please reformulate

A: suggestion followed.

Q:Fig 7 there are two panels a

A: suggestion followed.

Q:P17L359 what about H1=1000m or more, isn't it more realistic for the bottom layer of open ocean eddies?

A: Yes, the eddy thickness would be much thicker given weaker stratification in deeper layer of open ocean. If H1=1000m or more, then the upper and low layers are relatively smaller in two-layer model. So it is more like a plane model (as assumed in previous theoretic studies) but two-layer model in this study. Besides, the eddies in this region were only about a few hundred meters according to the previous statistics [e.g., Wang et al., 2017]. As it was illustrated in the sensitive analysis, this would not change the main results and conclusions.

[Figure]

Q:P18L387 in 2015 during a merger event.

A: suggestion followed.

Q:P18L392 be the sum of

A: suggestion followed.